# Fundamental Properties and Thermal Transferability of Masonry Built by Autoclaved Aerated Concrete Self-Insulation Blocks

**DOI:** 10.3390/ma13071680

**Published:** 2020-04-03

**Authors:** Fenglan Li, Gonglian Chen, Yunyun Zhang, Yongchang Hao, Zhengkai Si

**Affiliations:** 1School of Civil Engineering and Communication, North China University of Water Resources and Electric Power, Huayuan Campus, No. 36 Beihuan Road, Zhengzhou 450045, China; chengonglian@ncwu.edu.cn; 2International Joint Research Lab for Eco-building Materials and Engineering of Henan, North China University of Water Resources and Electric Power, Zhengzhou 450045, China; zhangyyun1990@163.com (Y.Z.); silence_is_gold666@163.com (Y.H.); 3Henan Xing’an New Building Materials CO., LTD, Xingyang National Electric Industrial Park, Zhengzhou 450000; China; sizhengkai@hnxan.com

**Keywords:** autoclave aerated concrete (AAC), self-insulation block, masonry, fundamental property, thermal conductivity, heat transfer coefficient

## Abstract

This paper performed a detailed study on the fundamental properties and thermal conductivity of autoclaved aerated concrete (AAC) self-insulation block, and the mechanical properties and heat transfer resistance of the AAC self-insulation block masonry. Different kinds of joints and the plastering surface were used to build the masonry specimens. The distinctive feature of the blocks and mortars is the lower thermal conductivity with expected strength. Compared to those with larger thickness of insulation mortar joints, the masonry with thin-layer mortar joints had better compressive performance and lower shear strength. The compressive strength of masonry was related with the block and mortar strengths, the shear strength of masonry along mortar joints was related with the mortar strength. The stress–strain relationship of masonry in compression could be predicted by the similar expression of conventional block masonry. The tested heat transfer coefficient of AAC self-insulation block masonry with thickness of 250 mm without plastering surfaces was (0.558 ± 0.003) W/(m^2^·K). With the plastering surfaces, the heat transfer coefficient reduced by 4.4% to 8.9%. Good agreements in values of heat transfer coefficient existed by using the test, theoretical computation and ANSYS (ANSYS Inc. Canonsburg, PA, USA) analytical methods. Based on the extensibility analyses, the heat transfer coefficients of AAC self-insultation block masonry with different thickness are proposed. The best thickness is proposed for the outer walls of residential buildings in different cold zone to meet the design requirement of energy conservation.

## 1. Introduction

In the past century, research and production technology of autoclaved aerated concrete (AAC) have been constantly developed. With the feature of integrated performance in thermal insulation, fireproofing and durability, AAC becomes a green lightweight building material [1,2,3]. In China, to meet the design standard of the 65% energy-saving buildings with class-A fireproof ability, AAC has popularly applied to construct the self-supporting walls as partition or filled in the frame structures and the bearing walls of low-rise civil buildings [4,5,6]. 

Based on the literature, a variety of important researches on AAC have performed to investigate the composition and microstructures affected by the constituent materials, quartz particle size, autoclaving process and the dry–wet circulation of service environment. The basic performances including density, water absorption, strength, drying shrinkage, freeze–thaw resistance as well as the thermal insulation properties were also studied [1,3,7,8,9,10,11,12,13,14,15]. AAC is commonly produced through molding and autoclaved aeration process of constituent materials including cement, lime, sand, water and aluminum powder. At present with the needs of environmental protection, the raw materials changed to be various industrial by-products and other wastes mainly containing silicon and calcium [3,6]. The industrial by-products included fly ash, silica fume, bottom ash and iron ore tailing, copper tailing, blast furnace slag, coal gauge [16,17,18,19]. A study indicated that using rice husk ash as partial replacement for fine aggregate reduced the strength and density of AAC with a tendency to reduce the autoclaving time and autoclaving temperature required [20]. Another effort was done to enhance the mechanical properties especially fracture toughness and promote the thermal insulation performance of AAC by using the wood fiber and the rubber powder [21]. The thermal conductivity of AAC was linear with the substitution of sand with equal volume fibers, the higher thermal conductivity got from the substituting of basalt fiber, the best compressive and flexural strengths got due to the reinforcement of carbon fiber [22]. Whatever the constituent materials used, the core concern is the thermal insulation of AAC with the premise of sufficient strength and durability. 

Normally, the thermal conductivity of AAC block varies from 0.1 to 0.7 W/(m·K) for density of 400–1700 kg/m^3^ [3]. In China, the thermal conductivity of AAC block applied in civil building varies from 0.13 to 0.22 W/(m·K) for density of 400–700 kg/m^3^ [6]. As per China codes [6,23,24], the thickness of the outer walls built by AAC blocks should be no less than 400 mm and 300 mm, respectively for residential buildings in the severe cold zone and the cold zone. This faces a challenge to reduce the thermal conductivity of AAC blocks to reduce the wall thickness. An effective method is filling AAC with phase changed materials [25,26]. This paper follows another route of improving the production technology to get the AAC with a lower thermal conductivity. To identify this new AAC, it is called as the AAC self-insulation block.

As an indispensable component of the masonry, the mortar joints play an important role in the loading and thermal transferring [27,28]. Normally, the loading performance of AAC block masonry is enhanced by the mortar joints due to the higher strength of mortar, while the thermal resistance of AAC block masonry is reduced due to the formation of a microthermal bridge through the mortar joints. Based on a case study, the energy consumption of masonry with insulation mortar could reduce by 21.5% [29]. The tensile and shear strength of masonry are greatly affected by the bond strength of block-mortar interface formed with different mortars [29]. Therefore, it is valuable to develop the thermal insulation masonry mortar. By replacing the river sand with lightweight sand, the thermal conductivity of mortar reduced by 25.9% to 46.5% [30]. By using ordinary silicate cement, fly ash, river sand, water retaining agent and dispersed latex powder, a thin-layer masonry mortar with thickness of mortar joint from 3 mm to 5 mm was developed [31]. However, the validity of the mortars used for the masonry with AAC self-insulation blocks needs to be further studied. 

At present, based on investigations of mechanical properties and thermal insulation performance of conventional AAC block masonry, the masonry structures can be designed in accordance with the standard specifications [6,27,28,29,32,33,34]. A composite wall with the AAC block masonry filled in and completely linked with the lightweight concrete frame was proposed by Gao et al. [35]. This provides a feasible technical solution for the loading combination in relation with the thermal insulation of AAC block masonry with lightweight concrete. A new attempt is to apply AAC in the seismic and energy retrofit of reinforced concrete framed buildings by the replacement of the external layer of double-leaf infill walls made of hollow bricks. With the improvement of seismic resistance, the energy demand can be reduced by 10% and 4% for heating and cooling, respectively [36]. However, in view of the thermal transfer properties of AAC block masonry, less experimental studies except numerical analyses were performed [27,37,38]. 

## 2. Research Significance

The AAC self-insulation block with a smaller thermal conductivity and the special masonry mortar were developed to meet the requirement of building energy conservation. However, as a new product, there is lack of the research on the fundamental properties of the block masonry in cooperation with the heat transfer property [6,32]. This is not conductive to the promotion of AAC self-insulation blocks due to no reliable data in standards. To bridge this gap, a detailed experimental study was conducted to determine the fundamental properties and the thermal transferability of the AAC self-insulation block, masonry mortars, plastering mortar and masonry. Test results are presented in this paper. Meanwhile, the design parameters and predictive formulas for the compressive strength and stress–strain relationship, the shear strength along mortar joints, and the heat transfer coefficient of AAC self-insulation block masonry are proposed.

## 3. Experimental Work

### 3.1. Production of AAC Self-Insulation Block

The specific mix proportion for a batch of AAC self-insulation blocks was 2900 kg pulverized coal ash slurry with a diffusion degree at 390–400 mm, 100 kg plaster paste, 330 kg lime, 260 kg ordinary silicate cement of 42.5 strength grade, 3.5 kg aluminum paste and 800 mL foam stabilizer. 

To produce AAC self-insulation blocks, the constitute materials are firstly prepared. The lime is prehomogenized with complete internal reaction for a period of time after ground, and the aluminum paste is dissolved in water at room temperature to be aluminum liquid. Then the slurry is made with a diffusion degree of 280 mm at temperature of 40.2 °C. The slurry is cast into the mould preheated at temperature of 50 °C and stands for 31 min at temperature of 64 °C to complete the process of foaming, thickening and hardening. After demolding, the blocks are placed into the autoclave for steam-curing for 3 h under the pressure of 1.0 MPa and the temperature of 185 °C. The blocks were 600 mm long, 300 mm height and 250 mm thick. As per the regulation of size deviation and appearance [5], the blocks belong to superior class product. 

### 3.2. Preparation of Mortars

Two kinds of masonry mortar were made of cement and sand by admixing water retaining agent and other additives. One was the thin-layer mortar used for the joints with thickness of 5 mm, in engineering it is commonly suitable for the superior product of blocks. The dry materials of thin-layer masonry mortar were market supplied [31]. The mass proportion of dry mortar materials to water was 1:0.48; the mixing time was about 5 min. Another was the insulation mortar used for the joints of qualified product of blocks with a thickness of 10 mm. The insulation mortar was made of thin-layer mortar admixed with expanded perlite and vitrified microsphere to reduce the heat loss of mortar joint. The mass proportion of dry mortar materials: expanded perlite and vitrified microsphere: water of 1:0.15:0.46, the mixing time was about 4 min.

The plastering mortar was prepared by admixing the polypropylene fiber with volume fraction of 0.04% into the thin-layer mortar. The function of polypropylene fiber is to prevent shrinkage cracking of plastering surfaces.

### 3.3. Tests for Fundamental Properties

The tests for fundamental properties of the AAC self-insulation blocks were in accordance with China code GB 11969 [39]. The standard cubic specimens with dimension of 100 mm were cut from the blocks. The dry density, moisture content, water absorption and compressive strength were measured. The loading direction was perpendicular to the curing steam direction of blocks. Tests were conducted on 100 kN hydraulic pressure tester with the loading speed of 0.15 kN/s. The tester was manufactured by Sansi Testing Machine and Equipment Co. Ltd., Tianjin, China.

The tests for fundamental properties of mortars were in accordance with China code JGJ/T 70 [40]. The dry density, segregation degree, compressive strength, bond strength, setting time at penetration resistance of 0.5 MPa, freeze-thaw resistance after 25 cycles and linear shrinkage were measured respectively by using the standard methods.

The tests for basic mechanical performances of AAC insulation block masonry were in accordance with China code GB/T 50129 [41]. As presented in Figure 1a, the compressive strength, deformation, crack pattern and failure modes of the masonry were measured. 

The axial deformation along the height and the transversal deformations at the middle height of the specimen were measured by two meters. Six masonry specimens, three per group, were built with the thin-layer mortar. Six masonry specimens, three per group, were built with the insulation mortar. As presented in Figure 1b, the shear strength along mortar joints, deformation, crack pattern and failure modes of the masonry were measured. The shear deformation at the middle height of the specimen was measured by a meter. Six masonry specimens, three per group, were built with the thin layer mortar. Six masonry specimens, three per group were built with the insulation mortar. 

The masonry specimens were covered by the plastic film for moisturizing and curing the mortar joints after construction. This remained for 28 days before testing.

### 3.4. Tests for Thermal Transferability

As per China code GB 10295 which equals to ISO 8301 [42], the one-dimensional steady-state heat flow method was used to measure the thermal conductivity of AAC self-insulation block and mortars. The heat flow meter was used as the test machine [15,30,43,44]. The plate specimens, two of them per trial, were 300 mm long and 300 mm wide, 50 mm thick for the block and 30 mm thick for mortars. The specimens of the block were cut from the block products. The specimens of mortars were prepared by casting in steel moulds.

As mentioned above, two kinds of mortar were used to construct the specimens of the block masonry. Three specimens per group were built respectively with the thin-layer mortar and the insulation mortar. The other group was built with the horizontal thin-layer mortar joints accompanied the vertical joints filled with elastic rock wool and sealed by the outside mortar. The product of elastic rock wool was market supplied with the dry density of 85 kg/m^3^, and the thermal conductivity of 0.12 W/(m·K). Figure 2 presents the construction of the block masonry specimens.

To ensure better bonding between the mortar and the blocks, the joint surfaces of the blocks were wetted by spraying water. The mortar joints were covered by plastic film for 28 days before testing. After tested for heat transfer coefficient, all specimens were plastered with the plastering mortar as displayed in Figure 2d. After maintaining for 28 days, the heat transfer coefficient of each specimen was measured again. 

Table 1 presents the details of AAC self-insulation block masonry specimens for the testing of thermal conductivity.

Of which the environmental relative humidity was the value at the time of starting test, the monthly environmental relative humidity at the curing stage of the specimens was 57% in May, 55% in June and 68% in July. The heat transfer performance of the specimens was measured during June and July.

The heat transfer coefficient of AAC self-insulation block masonry was measured by using WTRZ series testing machine produced by Shenyang Weite Co. Ltd., Liaoning, China. As per China code GB 10294, which is similar to ISO 8302 [45], the schematic diagram of the test method of the guarded hot plate apparatus is exhibited in Figure 3. 

The hot and cold boxes simulate the room condition and the external condition in winter, the measured area is 1.2 m × 1.2 m. The test machine and the installed specimen are exhibited in Figure 4. 

The specimen is installed in the mid-box between the hot and the cold, then the three boxes are tightened by the lock devices. After few hours of operation, the test system reaches steady state entirely, the air temperatures and surface airflow speeds on the two sides of specimen, the temperature of the protection chest can be measured. When the power *Q*_P_ of electric heater is inputted in the hot box, the heat quantity *Q*_1_ transmitted through the specimen can be measured. Finally, the heat transfer coefficient and total specific heat resistance of the specimen can be computed [46]. In this study, the temperature of hot and cold boxes was 30 °C and −10 °C.

## 4. Fundamental Properties

### 4.1. Fundamental Properties of AAC Block

The dry density, moisture content, water absorption and compressive strength of AAC blocks are presented in Table 2. 

The dry density of the block is 558 kg/m^3^, the moisture content is 1.3%, the water absorption rate is 63.5%, and the compressive strength is 4.0 MPa. The specimens presented higher brittleness during the loading process of compression. Visible cracks appeared when the load reached about 50% ultimate, then the cracks irregularly developed until the specimens damaged at the ultimate load. This outlines the lightweight feature of AAC self-insulation block with large porosity. The high-water absorption needs to take care of by covering water-proofing films before constructions.

### 4.2. Fundamental Properties of Mortars

The tested fundamental properties of masonry mortars and plastering mortar are presented in Table 3. 

The strength grade of masonry mortars fits the M7.5 as per China code GB 50003 [32] and is adapted to be applied for AAC self-insulation block masonry.

### 4.3. Compression Performance of AAC Block Masonry

As presented in Figure 5, three parts of the stress–strain curves reflect the loading performance of the AAC self-insulation block masonry in compression. 

The first linear part represents the elastic body of masonry before the short cracks initially appeared on the vertical mortar joint at 58%~75% peak stress. The second nonlinear part accompanies the extending of the initial crack up and down to the blocks until they went through the blocks. At the end, the stress reached the ultimate. After that is the third part: the cracks developed rapidly and divided the masonry into a plurality of column body, which crushed or became instable, then the whole masonry crushed as exhibited in Figure 6. 

Test results of featured values of the AAC block masonry under compression are presented in Table 4, of which the modulus of elasticity *E* is the secant modulus on stress–strain curve at a stress of 0.4*σ*_p_, and ε_0_ is the strain corresponding to the peak-stress *σ*_p_.

The Poisson’s ratio is computed by the tested transversal deformation. Due to the transversal expansion increased with the axial compressive stress, the Poisson’s ratio *ν* was taking in corresponding to the stress of 0.4*σ*_p_. 

Based on the modification of the formula of compressive strength for aerated concrete block masonry [6,32], the compressive strength *f*_c,m_ of AAC self-insulation block masonry can be calculated by Formula (1). The average ratio of tested to calculated compressive strength is 1.015, with a dispersion coefficient of 0.014.
(1)$$f_{c,m}=0.52f_{1}^{0.46}(1+0.07f_{2})$$ where *f*_c,m_ is the compressive strength of the masonry (MPa), *f*_1_ is the strength of the block (MPa), *f*_2_ is the strength of masonry mortar (MPa).

The modulus of elasticity of the block masonry with thin-layer mortar joints is 1.42 times of that with insulation mortar joints, and the Poisson’s ratio of the former is 1.13 times of the later. This is due to the direct connection of the blocks by thinner joints of thin-layer mortar. A slight effect of mortars was exerted due to the higher strength than that of AAC blocks.

Based on the stress–strain model proposed by Mander et al. [47,48], the stress–strain curve for the AAC self-insulation block masonry under uniaxial compression is expressed as follows:(2)$$\sigma =\frac{f_{c,m}\left(\frac{\varepsilon }{\varepsilon_{0}}\right)\gamma }{\gamma -1+{\left(\frac{\varepsilon }{\varepsilon_{0}}\right)}^{\gamma }}$$
(3)$$\gamma =E_{m}/\left(E_{m}-E_{\sec}\right)$$
where *E*_m_ is the average elastic modulus of the masonry, *E*_sec_ is the secant modulus with the ratio of *f*_c,m_ to *ε*_0_.

Another unified compressive stress–strain model used for all kinds of masonry in China code [32] is also applied for the AAC self-insulation block masonry, which is expressed as follows:(4)$$\varepsilon =-\frac{1}{\xi \sqrt{f_{c,m}}}\ln(1-\frac{\sigma }{f_{c,m}})$$ where *ξ* is the coefficient related to the elastic properties of masonry. 

The stress–strain curves of the block masonry computed by Formula (2) with average tested values of *f*_c,m_, *E*_m_ and *ε*_0_ are exhibited in Figure 5. Good fitness is obtained for the ascending portion of the curve, and better trends can be predicted although a large deviation exists at the descending portion. Meanwhile, Figure 5 also presents the calculation results of Formula (4) with *ξ* = 620; a greater difference exists for the masonry with thin-layer mortar joints after the stress is over 0.5*σ*_p_.

### 4.4. Shear Performance of AAC block Masonry

Generally, the masonry with higher shear strength has lower shear strain along the mortar joints. The AAC block masonry damaged in brittle under shear peak-load, most of them had cracks along the two sides mortar joints before the failure took place. Figure 7 presents two special failure modes of specimens under shear, of which specimens S1-3 and S2-5 failed along one side of the mortar joint accompanied with the bending fracture of the middle blocks. This led to the higher shear strength as presented in Table 5.

Opposite to the changes of compressive strength, the average shear strength of the masonry with thin-layer mortar joints is smaller than that with insulation mortar joints. Meanwhile, the dispersion coefficient of the former is smaller than that of the latter. This indicates the importance of good bond between mortar and blocks for the shear resistance. The block surfaces bonded well and remained entirety with a larger thickness of insulation mortar joints, which provided good condition of shear along the mortar joints. 

Based on the modification of the formula of shear strength for aerated concrete block masonry [32], the average shear strength of AAC self-insulation block masonry can be predicted as follows: (5)$$f_{v,m}=k_{5}\sqrt{f_{2}}$$
where *f*_v,m_ is the average shear strength of the masonry (MPa); *k* is the coefficient related to the mortar joints. *k* = 0.105 for the masonry with insulation mortar joints, and *k* = 0.084 for the masonry with thin-layer mortar joints.

## 5. Thermal Transferability

### 5.1. Test Results

The test results of heat transfer coefficient of the masonry are presented in Table 6. 

Without the plastering surfaces, the AAC self-insulation block masonry had the heat transfer coefficient of (0.558 ± 0.003) W/(m^2^·K). This indicates the slight influence of different mortar joints on the thermal insulation of the masonry. With the plastering surfaces, the heat transfer coefficient of the masonry reduced by 4.4% to 8.9%. This indicates that the plastering mortar benefits to rise the thermal resistance of the masonry. Moreover, the results provide another foundation of masonry construction having similar thermal insulation. This provides a selection of the convenient and fast construction for AAC self-insulation block masonry with the thin-layer mortar joints, the insulation mortar joints or the horizontal mortar joints with vertical joints filled by rock wool.

### 5.2. Theoretical Computed Results

Without the plastering surfaces, the AAC self-insulation block masonry composed with blocks and mortar joints, as presented in Figure 8. 

The thermal resistance is calculated by parallel mode due to the micro-thermal bridge effect of mortar joints [2,49]. Taking the blocks, mortar joints and rock wool as the single-layer homogeneous materials, the heat transfer resistances *R* can be computed as,
(6)R=δ/λ
where *δ* is the layer thickness of material (m); *λ* is the thermal conductivity of material [W/(m·K)].

The heat transfer resistance *R*_0_ of the masonry can be calculated as [2,49],
(7)$$R_{0}=A/\left(\frac{A_{\mathrm{jh}}}{R_{\mathrm{jh}}}+\frac{A_{\mathrm{jv}}}{R_{\mathrm{jv}}}+\frac{A_{b}}{R_{b}}\right)$$
where *R*_jh_, *R*_jv_, *R*_b_ are the heat transfer resistance of horizontal joints, vertical joints and blocks, respectively (m^2^·K/W); *A*_jh_, *A*_jv_, *A*_b_ are the area of horizontal mortar joints, vertical mortar joints and blocks, respectively (m^2^); *A* is the total area of the masonry (m^2^), *A* = *A*_jh_ + *A*_jv_ + *A*_b_.

With the plastering surfaces, the plaster layers are regarded as the layered homogeneous material, the heat transfer resistance *R*_p_ is also computed by Formula (5). The heat transfer resistance of the masonry with plastering is calculated by a series model as follows [2,49]
(8)$$R_{0}=R_{\mathrm{in}}+R_{t}+R_{p}+R_{\mathrm{ex}}$$
where *R*_in_, *R*_ex_ are the heat transfer resistance of internal surface and external surface. 

The heat transfer coefficient *K*_0_ [W/(m^2^·K)] of the masonry is calculated as,
(9)$$K_{0}=1/R_{0}$$

Under dry condition, the tested thermal conductivity of the self-insulating block, the insulation mortar, the thin-layer mortar and the plaster mortar was 0.11 W/(m·K), 0.19 W/(m·K), 0.50 W/(m·K) and 0.50 W/(m·K), respectively. However, the thermal conductivity of AAC block increases with the increase of moist content [15,44,50], the increment is about 0.02 W/(K·m) for AAC with dry density of 350–450 kg/m^3^ when the moist content is within 9% at environment temperature 20 °C and *RH* = 80% [44]; the increment is about 20% for AAC with dry density of 400–600 kg/m^3^ when the moist content is within 10% at environment temperature of 20 °C and *RH* = 20–40% [15]. Therefore, in the theoretical computations, the thermal conductivity of above materials was taken as 0.13 W/(m·K), 0.23 W/(m·K), 0.60 W/(m·K) and 0.58 W/(m·K), respectively.

Take *R*_in_ = 0.10 (m^2^·K/W) and *R*_ex_ = 0.01 (m^2^·K/W) based on the test data, the theoretical computed values of heat transfer coefficient are presented in Table 7. 

The heat transfer coefficient computed was almost lower than tested, and larger difference from 6.5% to 11.2% took place on the specimens without plastering while less difference from 1.7% to 8.8% took place on the specimens with plastering. This is due to the thermal bridge effect of mortar joints and rock wool with higher thermal conductivity than AAC self-insulation blocks. The exposed mortar joints had a higher thermal bridge than the plastered mortar joints. Accompanied with the uniform distribution of heat transferred from hot side surface through the plastering mortar, the thermal bridge effect can be weakened by the plastering layer. Meanwhile, the largest difference between tested and computed heat transfer coefficient took place on the specimens with vertical joints filled by rock wool. This may be due to the filling quality in this test only relied on the compression from the blocks on the two sides of mortar joint. If the rock wool is bonded to the blocks by adhesive, the thermal bridge effect of rock wool should be reduced.

### 5.3. Analytical Values by ANSYS

The two-dimensional analysis method is often used to analyze the thermal properties of walls, ignoring the influence of thermal conductivity of vertical mortar joints on the heat transfer coefficient of masonry wall. By using the ANSYS (ANSYS Inc. Canonsburg, PA, USA) 3D thermal analysis technology, the influence of horizontal and vertical mortar joints on heat transfer resistance of masonry wall can be comprehensively dealt with. The steady state heat transfer model of the masonry wall is built on the hypotheses of one-dimensional variation of temperature and symmetrical heat transfer. 

The blocks and mortar joints are divided into three-dimensional 8-node four-side SOLID70 elements, each node has one degree of temperature freedom. According to the energy conservation principle, the initial conditions and the boundary conditions, the heat balance equation is built for each node of SOLID70 element. Therefore, the temperature, heat flux gradient and heat flux density of each node are obtained.

In the ANSYS model, the tested temperature of hot and cold sides at 30 °C and −10 °C, and the tested thermal conductivity of AAC self-insulating block and insulation mortar, thin-layer mortar, plaster mortar of 0.13 W/(m·K), 0.23 W/(m·K), 0.60 W/(m·K) and 0.58 W/(m·K) were taken into account. The convective heat transfer coefficients at high and low temperatures were 9 W/(m^2^·K) and 25 W/(m^2^·K). The distribution nephograms of heat flux density and temperature gradient can be got. The average surface temperature and heat flux of the high and low temperature sides can be calculated by extracting their data of nodes. Then the heat transfer coefficient can be calculated by the average heat flux divided by the difference of average surface temperature between the two sides. Taking M1= specimen as an example, the average surface temperatures at high and low temperature sides are 27.63 °C and −9.13 °C, the average heat flux is 20.40 W/m^2^, then the heat transfer coefficient is 0.555 W/(m^2^·K). Similarly, the heat transfer coefficient of the masonry under different conditions can be obtained.

The ANSYS analytical results of heat transfer coefficient are presented in Table 7. Compared with the experimental, good agreement with average ratio of 1.002 is given out with slight differences. This provides a reliable method by using the ANSYS simulation to get the thermal transfer property of AAC self-insulation block masonry. 

## 6. Discussion

### 6.1. Loading Capactity of Masonry

The AAC self-insulation blocks had the compressive strength of 4.0 MPa, the dry density of 558 kg/m^3^, the moisture content of 1.3%, the water absorption rate of 63.5%. Based on the specification in China code GB 11968 [5], the block belongs to the superior product with class of strength A5.0 and density B06. The lower thermal conductivity of 0.11 W/(m·K) than the limit of 0.16 W/(m·K) provides a salient feature of the block masonry with higher thermal resistance. 

With rational compressive stress–strain relationship, the AAC self-insulation block masonry has compressive strength relied on the strengths of block and mortar, and the shear strength along mortar joints relied on the mortar strength. Without any doubt, the AAC self-insulation block masonry has enough self-supporting capacity and is adaptable to be used for the bearing walls of masonry structure buildings. The loading performances of AAC self-insulation block masonry under compression and shear along joints are similar with those of block masonry specified in design code for masonry structures [6,32].

### 6.2. Thermal Trasfer of Masonry

No obvious difference of heat transfer coefficient exists on the AAC self-insulation block masonry with three kinds of joints. This gives a selection of joints for the convenient constructions. The joints can be made with the insulation mortar and the thin-layer mortar, or with the horizontal mortar joint with vertical rock wool joint. 

The study provides the heat transfer coefficients of AAC self-insulation block masonry without plastering. This is to verify the function of plastering mortar on the thermal resistance. Generally, the test specimens with plastering had a reduced heat transfer coefficient about 5%. In view of that the plastering is part of the construction, the heat transfer coefficient of AAC self-insulation block masonry with plastering is proposed for building design of energy conservation.

### 6.3. Thermal Design of Masonry Walls

The extensibility analyses are performed to get the heat transfer coefficient of AAC self-insulation block masonry with different thickness of blocks. The plastering mortar is 5 mm thick. The blocks are produced with standard thickness of 150 mm, 180 mm, 200 mm, 250mm and 300 mm. The walls with thickness of 360 mm and 400 mm are composited by the blocks with thickness of 180 mm and 200 mm. Results of proposed heat transfer coefficient are presented in Table 7. 

The validity of AAC self-insulation block masonry used for the outer wall of residential buildings to meet the energy conservation of China codes JGJ 26 [23] and JGJ 134 [24] is also indicated in Table 7. 

The best thickness of the blocks applied for the outer wall of residential buildings with different storeys in different energy conservation zones are proposed. Generally, the AAC self-insulation block masonry can be applied for almost all the residential buildings except those with less of 3 storeys in the severe cold zones A and B. The AAC self-insulation block masonry with thickness no less than 300 mm can be applied for the outer wall of residential buildings with no less than 9 storeys. 

## 7. Conclusions

Based on the study of this paper, conclusions can be drawn as follows:(1)The superior products of the AAC self-insulation block were prepared with the compressive strength of 4.0 MPa and the dry density of 558 kg/m^3^. The AAC self-insulation block had a thermal conductivity of 0.11 W/(m·K). This is smaller than the limit of 0.16 W/(m·K) of conventional AAC block with similar strength and density.(2)Two kinds of masonry mortar and a plastering mortar were prepared for the AAC self-insulation block masonry. The fundamental properties of them were measured by experiments. The feature of the mortars is the lower thermal conductivity with expected strength. The thermal conductivity of the thin-layer mortar and the insulation mortar was 0.50 W/(m·K) and 0.19 W/(m·K), respectively. The thermal conductivity of plastering mortar was 0.48 W/(m·K).(3)The loading performances of AAC self-insulation block masonry under compression and shear along mortar joints are like that of the masonry with conventional blocks. The compressive strength of AAC self-insulation block masonry can be computed by the formula with the strengths of the block and mortar. The shear strength along mortar joints can be computed by the formula in relation with the mortar strength.(4)The heat transfer coefficient of AAC self-insulation block masonry can be determined with good agreement by the tests, the theoretical computation and the ANSYS analysis. A little deviation exists due to the difference of mortar joints including the thin-layer mortar joints, the insulation mortar joints and the horizontal mortar joints with filled vertical joints by rock wool. Therefore, the masonry can be built by any of each kind of mortar joints for the convenience of construction.(5)The heat transfer coefficient of AAC self-insulation block masonry wall with block thickness differing from 150 mm to 400 mm are proposed. Compared with the specification of China codes, the best thickness of AAC self-insulation block masonry is suggested for the outer wall of residential buildings in the severe cold zone, the cold zone and the hot summer and cold winter zone.

## Figures and Tables

**Figure 1 materials-13-01680-f001:**
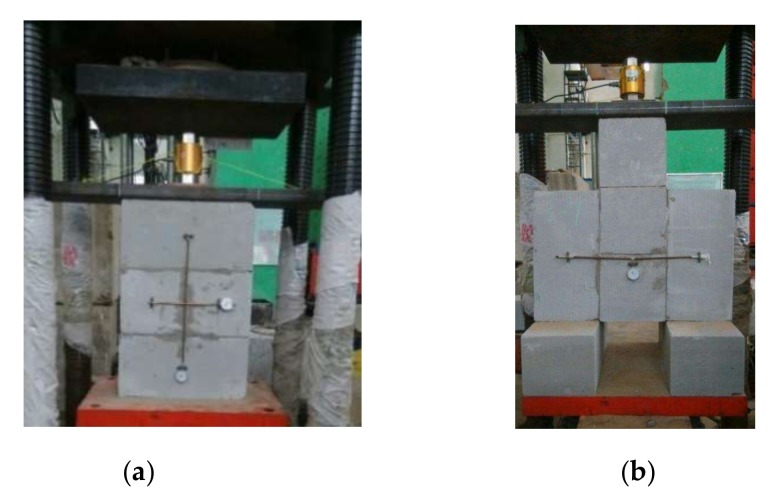
Tests of block masonry: (**a**) compression; (**b**) shear along mortar joints.

**Figure 2 materials-13-01680-f002:**
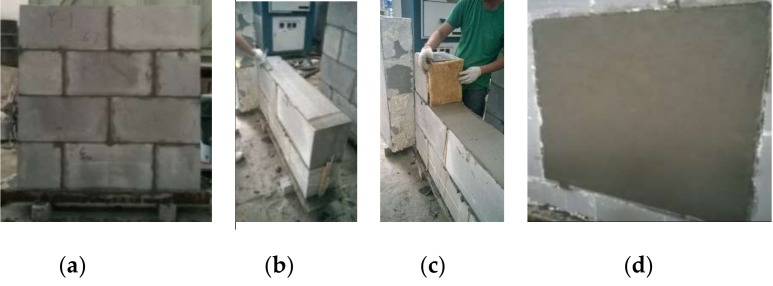
Construction of masonry specimens: (**a**) finished specimens; (**b**) flatting horizontal mortar joint; (**c**) rock wool for vertical joint; (**d**) finished specimens with plastering surface.

**Figure 3 materials-13-01680-f003:**
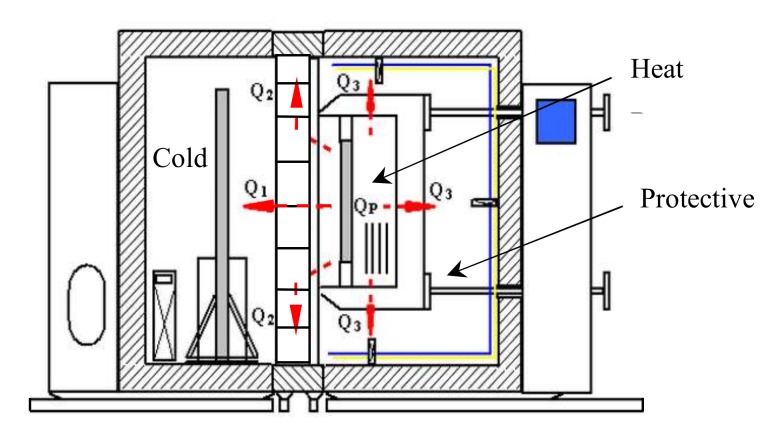
Testing schematic diagram for thermal conductivity.

**Figure 4 materials-13-01680-f004:**
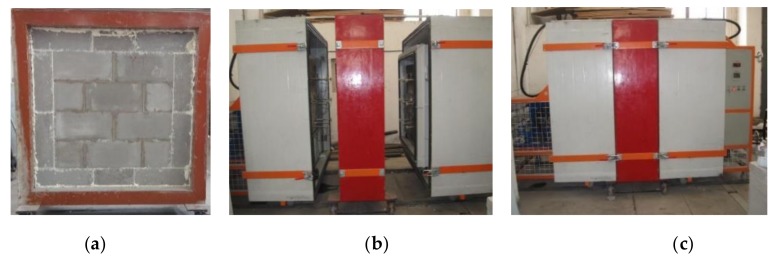
Testing device and installed specimen for thermal conductivity: (**a**) installed specimen; (**b**) internal meters; (**c**) close state of device.

**Figure 5 materials-13-01680-f005:**
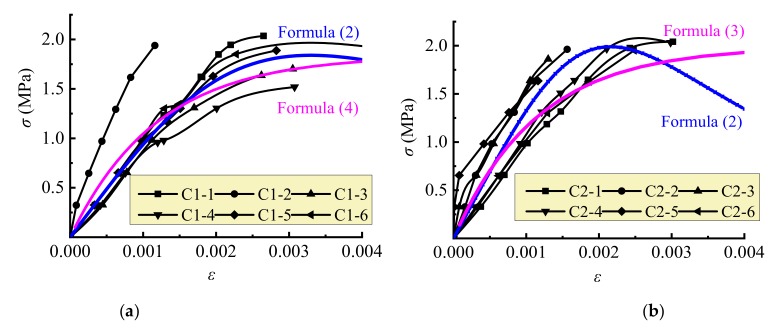
Compressive stress–strain curve of the block masonry: (**a**) insulation mortar joint; (**b**) thin-layer mortar joint.

**Figure 6 materials-13-01680-f006:**
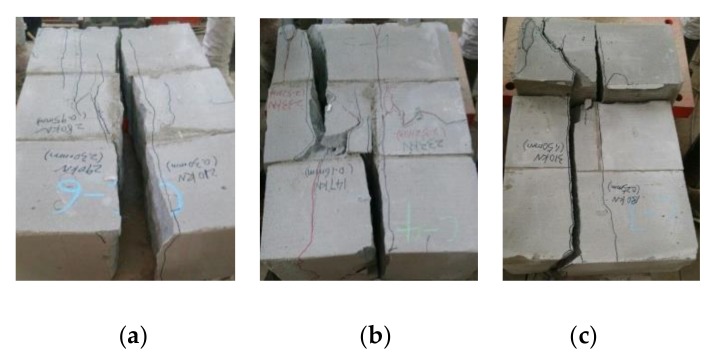
Compressive failure of the block masonry: (**a**) central crack; (**b**) unsymmetrical crack; (**c**) eccentric crack.

**Figure 7 materials-13-01680-f007:**
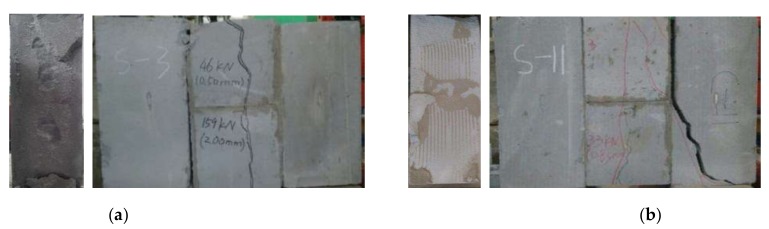
Shear failure pattern of AAC block masonry: (**a**) S1-3; (**b**) S2-5.

**Figure 8 materials-13-01680-f008:**
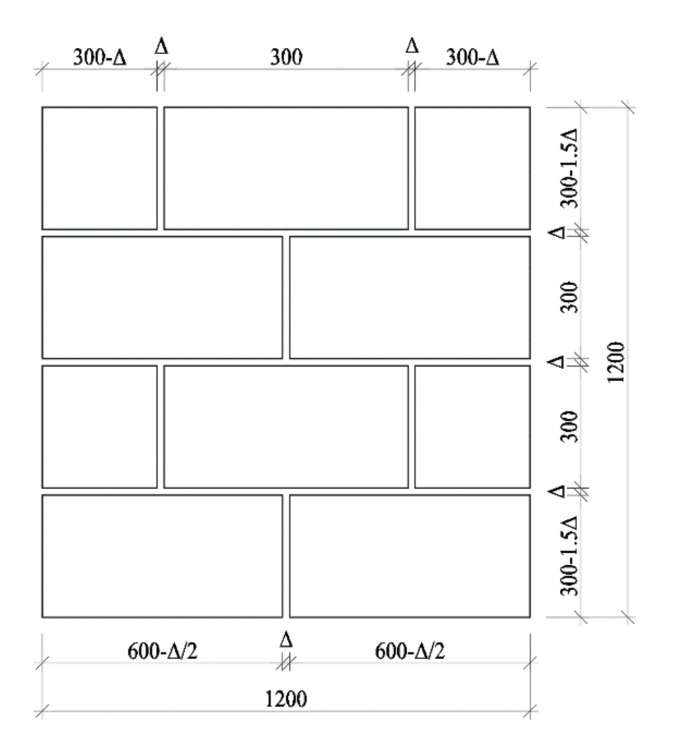
Composition of calculated masonry without plastering (unit: mm).

**Table 1 materials-13-01680-t001:** Details of AAC self-insulation block masonry specimens.

Group of Specimens	Wall Size (mm)			Mortar Joint (mm)		Plastering Thickness
	Length	Height	Width	Horizontal	Vertical	
M1	1205	1215	250	5	5	No
M1P	1205	1215	260	5	5	5
M2	1210	1230	250	10	10	No
M2P	1210	1230	270	10	10	10
M3	1205	1215	250	5	Rock wool	No
M3P	1205	1215	260	5	Rock wool	5

**Table 2 materials-13-01680-t002:** Test results of fundamental properties of AAC self-insulation blocks.

Specimen	Group 1	Group 2	Group 3
Dry density (kg/m^3^)	558	555	560
Moisture content (%)	1.43	1.07	1.33
Water absorption (%)	63.5	-	-
Cubic compressive strength (MPa)	3.9	4.1	4.1

**Table 3 materials-13-01680-t003:** Test results of fundamental properties of mortars.

Item	Thin-Layer Mortar	Insulation Mortar	Plastering Mortar
Dry density (kg/m^3^)	855	782	850
Segregation degree (mm)	10	12	9
Compressive strength (MPa)	13..6	12.0	9.3
Bond strength (MPa)	1.0	0.75	0.89
Setting time (h)	3.8	4.4	3.8
Freeze-thaw resistance after 25 cycles	Mass loss 2.9% Strength loss 12%	Mass loss 3.6% Strength loss 10%	Mass loss 2.5% Strength loss 11%
Linear shrinkage (mm/m)	0.70	0.90	0.65
Thermal conductivity [W/(m·K)]	0.50	0.19	0.48

**Table 4 materials-13-01680-t004:** Test results of compressive strength of AAC self-insulation masonry.

Specimen	*f*_1_ (MPa)	*f*_2_ (MPa)	*f*_c,i_ (MPa)	*E* (MPa)	*ν*	ε_0_	Mortar Joint
C1-1~C1-3	4.1	12.0	1.91	1021	0.158	0.00269	Insulation mortar with thickness of 10 mm
C1-4~C1-6	4.1	12.0	1.77	944	0.212	0.00398	
C2-1~C2-3	4.1	13.6	2.01	1628	0.243	0.00178	Thin-layer mortar with thickness of 5 mm
C2-4~C2-6	4.1	13.6	1.98	1925	0.157	0.00221	

**Table 5 materials-13-01680-t005:** Test results of shear strength of self-insulation AAC masonry.

Specimen	*f*_1_ (MPa)	*f*_2_ (MPa)	*f*_v,m_ (MPa)	Mortar Joint
S1-1~S1-3	4.1	12.0	0.390	Insulation mortar with thickness of 10 mm
S1-4~S1-6	4.1	12.0	0.343	
S2-1~S2-3	4.1	13.6	0.280	Thin-layer mortar with thickness of 5 mm
S2-4~S2-6	4.1	13.6	0.370	

**Table 6 materials-13-01680-t006:** Test results and analytical results for thermal transfer of AAC block masonry.

Group of Specimens	Measured Surface Thermal Resistance (m^2^·K/W)		Heat Transfer Coefficient [W/(m^2^·K)]		
	Hot Side	Cold Side	Tested Value	Theoretical Computed	ANSYS Analytical
M1	0.091	0.006	0.559	0.523	0.555
M1P	0.105	0.001	0.509	0.518	0.549
M2	0.096	0.006	0.561	0.505	0.548
M2P	0.087	0.012	0.533	0.497	0.529
M3	0.094	0.001	0.555	0.493	0.541
M3P	0.097	0.008	0.536	0.489	0.536

**Table 7 materials-13-01680-t007:** Proposed heat transfer coefficient of AAC self-insulation block walls and applied conditions.

Environment	Storeys of Building	Block Thickness (mm)	400	360	300	250	200	180	150
		*K*_0_ [W/(m^2^·K)]	0.35	0.40	0.48	0.55	0.66	0.72	0.84
Severe cold zone A	≤3	*K*_0_ ≤ 0.25	**×**	**×**	**×**	**×**	**×**	**×**	**×**
	4–8	*K*_0_ ≤ 0.40	**√**	**best**	**×**	**×**	**×**	**×**	**×**
	≥9	*K*_0_ ≤ 0.50	**√**	**√**	**best**	**×**	**×**	**×**	**×**
Severe cold zone B	≤3	*K*_0_ ≤ 0.30	**×**	**×**	**×**	**×**	**×**	**×**	**×**
	4–8	*K*_0_ ≤ 0.45	**√**	**best**	**×**	**×**	**×**	**×**	**×**
	≥9	*K*_0_ ≤ 0.55	**√**	**√**	**√**	**best**	**×**	**×**	**×**
Severe cold zone C	≤3	*K*_0_ ≤ 0.35	**√**	**×**	**×**	**×**	**×**	**×**	**×**
	4–8	*K*_0_ ≤ 0.50	**√**	**√**	**best**	**×**	**×**	**×**	**×**
	≥9	*K*_0_ ≤ 0.60	**√**	**√**	**√**	**best**	**×**	**×**	**×**
Cold zone	≤3	*K*_0_ ≤ 0.45	**√**	**best**	**×**	**×**	**×**	**×**	**×**
	4–8	*K*_0_ ≤ 0.60	**√**	**√**	**√**	**best**	**×**	**×**	**×**
	≥ 9	*K*_0_ ≤ 0.70	**√**	**√**	**√**	**√**	**best**	**×**	**×**
Hot summer and cold winter zone	all	*K*_0_ ≤ 1.0	**√**	**√**	**√**	**√**	**√**	**√**	**best**

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
