# Peer review of "Fundamental Properties and Thermal Transferability of Masonry Built by Autoclaved Aerated Concrete Self-Insulation Blocks"

_materials, 2020, doi:10.3390/ma13071680_

Round 1

Reviewer 1 Report

  1. Abstract should be overwrite, be giving shortly important information.
  2. The introduction and most reference are outdate, mostly older than 5 years. Also authors do not use new literature sources from “Materials”. The match to journal scope is questionable.

Author Response

Dear Professor,

Thanks very much for reviewing my paper. The response to your comments is presented as follow, please check them.

  1. Abstract should be overwrite, be giving shortly important information.

Reply: Ok, the abstract is revised as your comment.

  1. The introduction and most reference are outdate, mostly older than 5 years. Also authors do not use new literature sources from “Materials”. The match to journal scope is questionable.

Reply: The original references contain 11 papers published in recent 5 years. This is not best to reflect overall status of the research. In the revised manuscript, 10 papers published since 2016 including 3 from Materials are added as references, and the introduction is rewritten.

Best regards,

Fenglan Li

Reviewer 2 Report

Contrary to authors' words, the AAC is not a new generation material. It was invented almost 100 years ago, and claiming some novelty in its production and use seems to be exagerated.

Author Response

Dear Professor,

Thanks very much for reviewing my paper. The response to your comment is presented as follow, please check them.

Contrary to authors' words, the AAC is not a new generation material. It was invented almost 100 years ago, and claiming some novelty in its production and use seems to be exagerated.

Reply: Yes, the conventional AAC was invented almost 100 years ago. In this paper, we think the AAC with much lower thermal conductivity has some novelty by improving the production technology. So, we call this kind of AAC as self-insulation AAC. From the thermal insulation mechanism, the self-insulation AAC is promoted from the conventional AAC. To avoid the misunderstanding, we delete the claiming expression “a new generation material”.

Best regards,

Fenglan Li

Reviewer 3 Report

The English language is poor and requires a heavy revision.

Rows 67-68

Thermal conductivity data are attributed to [8], but none of these data are present in [8].

Rows 71

An effective method is filled AAC with phase changed materials.

“An effective method is filling AAC with phase changing materials”  ?

Rows 73

To identifier this new AAC

“To identify this new AAC”  ?

Rows 137

for the joint the joint with

?

Rows 212

in Figure 3, the test method of guarded hot plate apparatus was in accordance with China code GB 10294 which equals to ISO 8302

The apparatus in figure 3 is the “Guarded Hot Box” and it is described in ISO 8990 standard.

Rows 216

The calibration coefficient was 3.88.

What does it mean?

Table 8

Negative values for the Measured surface thermal resistance on the Cold side are non sense.

0.001 m2K/W means 1000 W/(m2K), how could you get that value?

Rows 403

by ANASYS

by ANSYS

Rows 409

one-dimensional freedom of temperature

one-dimensional variation of temperature

Rows 425

9 W/(K·m) and 25 W/(K·m)

9 W/(K·m2) and 25 W/(K·m2)

Rows 434

the average heat flux is 20.40 W/(K·m), then the heat transfer coefficient is 0.555.

the average heat flux is 20.40 W/m2, then the heat transfer coefficient is 0.555 W/(K·m2).

Rows 485

The superior products of AAC self-insulation block belong were prepared

The superior products of the AAC self-insulating block were prepared.

References [8] and [9] were published in 1995. Please correct the date.

Author Response

Dear Professor,

Thanks for reviewing my paper. We try our best to revise the English language, please check it.

The responses to your comments are presented as follow:

  • Rows 67-68

Thermal conductivity data are attributed to [8], but none of these data are present in [8].

 Reply: We did not present this data attributed to [8]. The reference No. is [3] in the manuscript.

  • Rows 71

An effective method is filled AAC with phase changed materials.

“An effective method is filling AAC with phase changing materials”?

 Reply: Yes, it has been revised.

  • Rows 73

To identifier this new AAC

“To identify this new AAC”?

 Reply: Yes, it has been revised.

  • Rows 137

for the joint the joint with?

 Reply: Sorry, it was repeated.

  • Rows 212

in Figure 3, the test method of guarded hot plate apparatus was in accordance with China code GB 10294 which equals to ISO 8302

The apparatus in figure 3 is the “Guarded Hot Box” and it is described in ISO 8990 standard.

 Reply: Thanks. The sentence has been corrected as: As per China code GB 10294 which equals to ISO 8302 [46], the schematic diagram of the test method of guarded hot plate apparatus is exhibited in Figure 3.”

  • Rows 216

The calibration coefficient was 3.88.

What does it mean?

  Reply: This is a coefficient to be used to calibrate the precision of test machine before formal testing. As an intermediate parameter of experiment, it has been deleted in the revised manuscript.

  • Table 8

Negative values for the Measured surface thermal resistance on the Cold side are non sense.

0.001 m2K/W means 1000 W/(m2K), how could you get that value?

 Reply: Yes, negative values are no sense. All data are recorded by the tester. We check the test data again and find it had slight influence on the test results of heat transfer coefficient. In the revised manuscript, the intermediate data are deleted, and the average values of each group are presented.

  • Rows 403

by ANASYS

by ANSYS

 Reply: Yes, by ANSYS.

  • Rows 409

one-dimensional freedom of temperature

one-dimensional variation of temperature

 Reply: Ok, it is more appropriate.

  • Rows 425

9 W/(K·m) and 25 W/(K·m)

9 W/(K·m2) and 25 W/(K·m2)

Reply: Thanks, they have been corrected.

  • Rows 434

the average heat flux is 20.40 W/(K·m), then the heat transfer coefficient is 0.555.

the average heat flux is 20.40 W/m2, then the heat transfer coefficient is 0.555 W/(K·m2).

 Reply: Thanks, they have been corrected.

  • Rows 485

The superior products of AAC self-insulation block belong were prepared

The superior products of the AAC self-insulating block were prepared.

 Reply: Thanks, I think “self-insulation” is ok.

  • References [8] and [9]were published in 1995. Please correct the date.

Reply: Ok, they have been corrected.

Thanks again,

Best regards,

Fenglan Li

Reviewer 4 Report

The research manuscript contains results of main physical and mechanical properties of autoclaved aerated concrete self-insulation block.  The first impression/comment of the Reviewer after reading the text is related to the formula of the entire manusctipt. It abounds in test descriptions and results for a specific construction product/mansonry. It is difficult for the Reviewer to determine what the novelty of the research covered in the text is. The manuscript is a very cross-sectional description of masonry built by autoclaved aerated concrete without deeper discussion on results. The text looks like a report on a tests on a construction product, with a relatively short discussion about the importance of individual results.

The conclusions in the manuscript are unsatisfactory. They contain de facto data on the results obtained in the research. According to the reviewer, based on these studies, several separate research problems could be described, provided that they were based on a separate discussion with the literature results. Of course, these results can be used by other researchers dealing with aerated concrete research, however, this study should put more specific research problem and focus on it.

Although the Reviewer does not feel competent to assess the linguistic correctness of work, he suggests linguistic correction. The sentences in the manuscript are very long, which often makes reading difficult. The authors are suggested to divide the sentences. For example, the first sentence in an abstract has 50 words, of which it contains the phrase "ACC self-insulation block" three times (one full ACC). (This is just one of many sentences that is difficult to read because of its length).

In terms of presentation of results - Reviewer does not see the need to attach the results of each one speciement. Perhaps some of the table contents could be added as an Appendix to this manuscript.

In summary, manuskrupt includes research on many of the physical and mechanical characteristics of ACC. Nevertheless, the work needs to be supplemented with content, especially in the "discussion" part (e.g. comparison with a large number of similar studies conducted by other scientists). Such a comparison will allow the formulation of scientific conclusions / solutions to the scientific problem.

The work also repeats information (Results-discussion-conclusions) about the results obtained, according to the Reviewer, not necessarily in a new approach. If we evaluate the quality of scientific research and manuscript as a research report on the material / construction technique for, for example, the Investor, the whole looks very good.

Finally, it is obvious - both the abstract and the whole text is too long. Perhaps it is worth adding that the same information could be contained 60% of the volume of the article.Line 53: Cement is not raw material.

Details: line 192 - self insulation;  ling 199: 28 days.

Author Response

Dear Professor,

Thanks for reviewing my paper. We try our best to strengthen the scientific discussion and the comparison with current design codes or the published papers to revise the manuscript. The responses to your comments are presented as follow:

  • The research manuscript contains results of main physical and mechanical properties of autoclaved aerated concrete self-insulation block.  The first impression/comment of the Reviewer after reading the text is related to the formula of the entire manusctipt. It abounds in test descriptions and results for a specific construction product/mansonry. It is difficult for the Reviewer to determine what the novelty of the research covered in the text is. The manuscript is a very cross-sectional description of masonry built by autoclaved aerated concrete without deeper discussion on results. The text looks like a report on a tests on a construction product, with a relatively short discussion about the importance of individual results.

Reply: Thanks. This paper presents the experimental results of AAC self-insulation block, the mortar and the masonry. The validity of design by using the current codes is analyzed. As a new product of AAC block with lower thermal conductivity, it is necessary to be studied before applying in engineering. You comment helps us to rewrite the paper highlighting the scientific content of this study. Please check it.

  • The conclusions in the manuscript are unsatisfactory. They contain de facto data on the results obtained in the research. According to the reviewer, based on these studies, several separate research problems could be described, provided that they were based on a separate discussion with the literature results. Of course, these results can be used by other researchers dealing with aerated concrete research, however, this study should put more specific research problem and focus on it.

Reply: The conclusion has been rewritten according to the compressing content of the revised paper.

  • Although the Reviewer does not feel competent to assess the linguistic correctness of work, he suggests linguistic correction. The sentences in the manuscript are very long, which often makes reading difficult. The authors are suggested to divide the sentences. For example, the first sentence in an abstract has 50 words, of which it contains the phrase "ACC self-insulation block" three times (one full ACC). (This is just one of many sentences that is difficult to read because of its length).

Reply: Thanks. The long sentences have been shortened or divided into several sentences.

  • In terms of presentation of results - Reviewer does not see the need to attach the results of each one speciement. Perhaps some of the table contents could be added as an Appendix to this manuscript.

Reply: Thanks. The intermediate data of experiments have been deleted. The final results are presented in the revised manuscript. Please check the Tables 2 to 7.

  • In summary, manuskrupt includes research on many of the physical and mechanical characteristics of ACC. Nevertheless, the work needs to be supplemented with content, especially in the "discussion" part (e.g. comparison with a large number of similar studies conducted by other scientists). Such a comparison will allow the formulation of scientific conclusions / solutions to the scientific problem.

Reply: Thanks. The comparisons of the experimental results to current design code or referenced papers are made in each part of the text. In the section of discussion, we want to give a proposal for the application of the block masonry in different cold zone, after comparison with the current design code. This provides a foundation to be referenced for the engineering design. According to your comment, we have strengthened the scientific discussion and comparisons in the revised manuscript.

  • The work also repeats information (Results-discussion-conclusions) about the results obtained, according to the Reviewer, not necessarily in a new approach. If we evaluate the quality of scientific research and manuscript as a research report on the material / construction technique for, for example, the Investor, the whole looks very good.

Reply: Thanks. We revise the manuscript to present the important results of the basic properties and thermal insulation of the AAC blocks. This block was produced to get a lower thermal transfer coefficient to get a better energy conservation of buildings. The test results are necessary for the design and construction.

  • Finally, it is obvious - both the abstract and the whole text is too long. Perhaps it is worth adding that the same information could be contained 60% of the volume of the article.

Reply: The paper has been shortened by 4 pages from 19 to 15 to present the main information of this study.

  • Line 53: Cement is not raw material.

Reply: “raw material” has been revised as “constituent materials”.

Details: line 192 - self insulation;  ling 199: 28 days.

Reply: Ok. They have been revised.

Thanks again,

Best regards,

Fenglan Li

Reviewer 5 Report

Replace or remove nephogram as it is not correct. 

I am not sure why rock wool would be used in the vertical joints as it is likley to reduce the overall structural integrity.

Author Response

Dear Professor,

Thanks for reviewing my paper. The responses to your comments are presented as follow:

  • Replace or remove nephogram as it is not correct. 

Reply: To compress the pages as per other reviewer’ comment, we remove this figure.

  • I am not sure why rock wool would be used in the vertical joints as it is likley to reduce the overall structural integrity.

Reply: This method is attempting to apply for the self-supporting wall filled in frames. It is convenient for construction than filled mortar in vertical joints.

Best regards,

Fenglan Li

Round 2

Reviewer 2 Report

As the AAC is quite well-known material I would put more emphasis on your attemps in changing its components, thus its properties. And then, any comparison to a "standard" AAC block would be much appreciated (as a table, for example, with some relative values).

And, by the way, what background was for calling AAC a "green" material (line 42)? Its components or production process? Are there any studies on the environment impact of AAC production? Take it as a question only, I don't expect instant answer in your paper.

Edit note: there is no table 1, so the rest should be renumbered.

Reviewer 3 Report

-

Reviewer 4 Report

Although the manuscript has been improved, I have the same remarks (pointed in the first review report).